# Environmental Contaminants Acting as Endocrine Disruptors Modulate Atherogenic Processes: New Risk Factors for Cardiovascular Diseases in Women?

**DOI:** 10.3390/biom12010044

**Published:** 2021-12-28

**Authors:** Silvia Migliaccio, Viviana M. Bimonte, Zein Mersini Besharat, Claudia Sabato, Andrea Lenzi, Clara Crescioli, Elisabetta Ferretti

**Affiliations:** 1Department of Movement, Human and Health Sciences, University of Rome Foro Italico, 00135 Rome, Italy; viviana.bimo@gmail.com (V.M.B.); clara.crescioli@uniroma4.it (C.C.); 2Department of Experimental Medicine, Sapienza University of Rome, 00161 Rome, Italy; zeinmersini.besharat@uniroma1.it (Z.M.B.); claudia.sabato@uniroma1.it (C.S.); andrea.lenzi@uniroma1.it (A.L.); elisabetta.ferretti@uniroma1.it (E.F.)

**Keywords:** endocrine disruptors, women, atherosclerosis, cadmium, bisphenol A, inflammatory cytokines, cardiovascular diseases

## Abstract

The number of aged individuals is increasing worldwide, rendering essential the comprehension of pathophysiological mechanisms of age-related alterations, which could facilitate the development of interventions contributing to “successful aging” and improving quality of life. Cardiovascular diseases (CVD) include pathologies affecting the heart or blood vessels, such as hypertension, peripheral artery disease and coronary heart disease. Indeed, age-associated modifications in body composition, hormonal, nutritional and metabolic factors, as well as a decline in physical activity are all involved in the increased risk of developing atherogenic alterations that raise the risk of CVD development. Several factors have been reported to play a role in the alterations observed in muscle and endothelial cells and that lead to increased CVD, such as genetic pattern, smoking and unhealthy lifestyle. Moreover, a difference in the risk of these diseases in women and men has been reported. Interestingly, in the past decades attention has been focused on a potential role of several pollutants that disrupt human health by interfering with hormonal pathways, and more specifically in non-communicable diseases such as obesity, diabetes and CVD. This review will focus on the potential alteration induced by Endocrine Disruptors (Eds) in the attempt to characterize a potential role in the cellular and molecular mechanisms involved in the atheromatous degeneration process and CVD progression.

## 1. Introduction

As the number of older individuals continues to increase, it is important to understand the pathophysiological mechanisms of age-related pathologies in order to develop interventions that can be easily implemented and contribute to “successful aging” and prevention of chronic diseases.

Age-related changes in body composition, metabolic factors, and hormonal levels, accompanied by a decline in physical activity, might all provide mechanisms responsible for the tendency to lose muscle mass, gain fat mass and develop cardiovascular diseases [1]. Indeed, cardiovascular diseases (CVD) are important widespread health problems that lead to a high prevalence of both mortality and morbidity, and during the past decades they have become a major health threat around the world [1,2,3].

Ageing also increases the risk of muscle mass reduction with a corresponding increase of fat mass and inflammation which, in association with hormonal imbalance and altered nutritional pattern [4], might synergistically increase CVD [3,5]. Of note, these age-related alterations are often sex-related as well [6,7,8].

Obesity, caused by an imbalance in which energy intake exceeds energy expenditure over a prolonged period, has always been recognized as a risk factor for metabolic disorders and CVD [9]. In particular, obese postmenopausal women are often affected by hypertension, dyslipidemia, diabetes mellitus and CVD, presenting an even higher risk compared to men [10].

In men, the condition of late-onset hypogonadism, frequently observed in the elderly, correlates with changes in body composition and increased cardiovascular risk [6,8,11,12]. Furthermore, recent epidemiological studies indicate that reduced androgen levels are correlated with high blood pressure, left ventricular mass and increased cardiovascular mortality in men [13].

Moreover, recent data indicate that some environmental pollutants, such as Cadmium (Cd) and Bisphenol A (BPA), which are widespread in the environment and can be introduced in the human organism in different ways, can cause significant alterations in human health, acting as endocrine disruptors (Eds). In particular, recent data suggest that the cardiovascular system might be a target of both of the pollutants Cd and BPA [14].

Thus, the aim of this review is to evaluate data on Eds, focusing on mechanisms of endothelial cell homeostasis disruption potentially leading to an increased risk of cardiovascular diseases, and addressing, when possible, sex-dependent differences.

## 2. Search Strategy

A systematic review of the literature was conducted using the following keywords: “Endocrine disruptors”, “Cardiovascular risk”, “Cadmium”, “Bisphenol A”, “Endothelium damage” and “Atherosclerosis” on the search engines “PubMed” and “Scopus”. Articles that were not related to the aim of this review and/or reported results using inappropriate cell models or a small number of patients were excluded. Two researchers, evaluating independently the titles and abstracts of the identified articles, performed the initial screening. A third evaluator was consulted when agreement between the two researchers could not be reached. The initial literature search identified a total of 1400 articles, and a manual screening of titles and keywords removed 610 records. After reviewing abstracts and full-text manuscripts, 100 papers fulfilling the above-mentioned search criteria were included. Figure 1 depicts the PRISMA flow diagram and the number of records included in the different phases of the review.

## 3. Pro-Inflammatory Cytokines Effects on Cardiovascular System

Atherosclerotic plaque can be considered the main expression of atherosclerosis, the main cause of CVD and the first cause of death among the population in industrialized countries. The atherosclerotic process starts from the endothelial cells, which are capable of processing many active substances and modulating the biological activity of the various vessel-wall structures, blood cells and proteins of the coagulation system, normally in contact with the surface of the endothelium [15]. When endothelium homeostasis is compromised by risk factors such as smoking, hypertension, obesity, diabetes and environmental stressors [16,17], this event might lead to upregulation of adhesion molecules, secretion of cytokines and chemokines and alteration of adhesion molecules [15].

Pro-inflammatory cytokines, which include several adipokines, are involved in many pathological processes, including inflammation, endothelial damage, atherosclerosis and hypertension. Their dysregulation is a strong contributing factor of the low-grade inflammatory state, which leads to a cascade of metabolic alterations inducing an increased risk of cardiovascular complications [18,19].

Tumor necrosis factor-alpha (TNF-α) is a pro-inflammatory cytokine that plays important regulatory effects on lipid metabolism, adipocyte function and insulin signaling [20]. In obese rats, TNF-α produced by periarteriolar fat alters endothelium-dependent vasodilatation, likely by inhibiting the insulin-mediated release of nitric oxide (NO) [21]. Moreover, recent results indicate that TNF-α upregulates the release of the adhesion molecules intercellular adhesion molecule 1 (ICAM-1) and vascular cell adhesion protein 1 (VCAM-1) in endothelial cells, facilitating leukocytes adhesion to vessel walls [22]. Thus, TNF-α may play an important role in vascular disease, confirming a pivotal role of this pro-inflammatory cytokine in the pathogenesis of atherosclerosis, endothelial damage and heart-cell remodelling toward higher disease severity [23].

Interleukin-6 (IL-6) is a cytokine that has a wide range of actions, including promotion of coagulation and immune/inflammatory reaction [24]. This cytokine is produced by different cell types, including endothelial cells; its levels can significantly increase, for instance, after menopause and with the decades of life, determining, along with increased levels of other cytokines, a subclinical chronic inflammatory status [25,26,27]. Interestingly, IL-6 has also been demonstrated to be an important correlation factor between inflammation and atherosclerosis since TNF-α can stimulate IL-6, which in turn can modulate C-reactive protein (CRP, an inflammatory biomarker of cardiovascular risk) production in smooth muscle cells, negatively affecting the expression of adhesion molecules and endothelial function [28]. Moreover, cohort studies have shown that increasing levels of this pro-inflammatory cytokine appear to be correlated with an increased risk (2-fold) of cardiovascular and all-cause mortality in healthy aged people, also having a significant prognostic value in subjects affected by unstable angina [29].

Angiotensin (AT), predominantly produced by the liver and adipose tissue, is the precursor of the vasoactive peptide angiotensin II and it appears correlated to higher blood pressure [30].

Plasminogen activating inhibitor (PAI-1), produced by liver and adipose tissue, inhibits the activity of tissue-plasminogen activator favouring thrombus formation over ruptured atherosclerotic plaques. PAI-1 expression is elevated in visceral obesity, insulin resistance (IR) and hypertriglyceridemia, and its levels appears to predict risk for future development of both type 2 diabetes (T2D) and CVD [31].

Leptin, the first identified adipose tissue-derived factor, is secreted by adipocytes in proportion to body fat tissue. Interestingly, hyperleptinemia, often present in subjects affected by overweight or obesity, has been widely recognized as an independent cardiovascular risk factor [32,33]. Several data suggest that hyperleptinemia might play a pivotal role in the pathogenesis of endothelial dysfunction and atherogenesis, likely stimulating the release of oxygen reactive species (ROS) as well as the recruitment of monocytes [33]. Leptin induces macrophage cholesterol ester synthesis, contributing to foam cell formation in vitro [34] with high glucose levels, also inducing the expression of CRP [35].

Resistin is produced by macrophages and visceral adipocytes, and its name derives from the induction of IR [36]. Resistin modulates insulin sensitivity in both skeletal muscle and liver, and positively correlates with IR and glucose tolerance in both human and animal models [37]. Resistin is believed to be a marker of inflammation, contributing to atherogenesis. Indeed, in vitro data obtained in human endothelial cells show that resistin induces a dose-dependent proliferation of smooth muscle cells and increases endothelin-I release, VCAM and ICAM-1 [38,39]. In addition, resistin appears to be a good predictor marker of coronary artery calcification [40], being also associated with arterial stiffness [41], while it seems inversely associated to left ventricular fractional shortening, biomarker of left ventricular systolic function [42]. Recent evidence indicates how resistin is independently linked with an increase in the risk of both myocardial infarction and ischemic stroke [43]. Table 1 summarises the cytokines involved in cardiovascular disorders.

## 4. Endocrine Disruptors

The term endocrine disruptors (Eds) implies several chemicals, with a particular effect on the endocrine system, since they interfere with specific receptor-mediated hormone activity [44]. Due to this characteristic, Eds can alter cellular metabolism with potential long-term and harmful effects. Eds are molecules of either natural origin or man-made products, which include over 300 synthetic compounds that includes chemicals such as the plasticizers polybrominateddiphenyl ethers (PBDEs) and polychlorinated biphenyl (PCB); insecticides (i.e., dichlorodiphenyltrichloroethane DDT and metabolites, pyrethroids); herbicides (i.e., atrazine, nitrofen); fungicides (i.e., zineb, ziram); pharmacological agents (i.e., bisphenol A—(BPA)) [45,46,47,48,49,50]; dioxins; dioxin-like compounds; phthalates; and heavy metals such as lead, mercury and cadmium (Cd) [45]. Due to this distinctiveness, there is a rising concern about how the endocrine or cardiovascular systems are affected by Eds, such as Cd or BPA, since it has been demonstrated that these molecules might mimic the activity of natural hormones such as estrogens and androgens, leading to the activation of specific signaling pathways [51]. Of note, Eds can block the interaction of these hormones with their natural receptors [52,53] or enhance the levels of proinflammatory cytokines [54].

## 5. Endocrine Disruptors and Cardiovascular System

As already mentioned above, CVDs are disorders that affect the blood vessels and heart, representing one of the leading causes of both morbidity and mortality worldwide. Risk factors for CVD include unhealthy diet [11], sedentary lifestyle, alcohol abuse, smoke and pollution [55]. For instance, some pollutants acting as Eds have been correlated to an increased risk of developing CVD due to a direct and specific alteration in pro-inflammatory cytokine levels and endothelium damage, leading to atherosclerotic lesions. Cd and BPA, two Eds that have been highly correlated with CVD, will be further described.

### 5.1. Cadmium and Cardiovascular Effect

Cd is a toxic heavy metal, found in soil, contaminated water and food, that is used in various industrial activities, and a non-occupational source is represented by cigarette smoking, as Cd accumulates in tobacco leaves. Several studies indicate a negative effect of this Ed on CVD. The molecular mechanisms by which Cd exerts the negative effects on the cardiovascular tissues are linked to the induction of oxidative stress, since it might disrupt endogenous antioxidant defence such as glutathione peroxidase (GPx), catalase (CAT) and superoxide dismutase (SOD). In addition, Cd induces ROS generation [56], harms the mitochondrial electron chain transport and decreases the antioxidant scavengers such as glutathione (GSH), leading to an imbalance in the cellular redox state and, so far, triggering the production of ROS [57,58,59].

#### 5.1.1. Clinical Studies

Clinical studies have indicated that this heavy metal acts as a pro-atherogenic factor since its presence has been identified in carotid plaques, leading to a significant increase in vulnerability of the plaques compared to plaques that do not fissure and rupture [60,61,62]. In detail, epidemiological studies showed that a high serum level of Cd was linked with CVD mortality and carotid plaques’ prevalence in a Swedish population and was also correlated with an increase in CVD risk in the Korean male population [63,64]. Moreover, a follow-up study performed for almost 20 years on a Swedish population-based cohort of over 4000 middle-aged subjects of both sexes demonstrated that Cd might play a pivotal role in smoking-induced CVDs, by measuring the level of Cd in the blood [65].

Another interesting study demonstrated a correlation between high urine and blood concentration of Cd and plaque formation in a female and male population over 60 years of age [66], indicating that even if Cd likely acts by disrupting the estrogen receptor pathway, both genders are affected by the pollutants’ negative action on cardiovascular health.

Interestingly, several studies reported that Cd accumulation correlates with increased macrophages presence, a recognized hallmark of symptomatic and vulnerable carotid plaques [67,68]. In detail, recently published data obtained from a Canadian population indicated a correlation between pollutants and carotid intima-media thickness (CIMT) [69]. The hypothesis that Cd triggers the vulnerability of carotid plaques, likely by increasing the risk of rupture and ischemic stroke, was supported by a recent study that showed that Cd accumulation was linked to the incidence of ischemic stroke [62].

It is well known that cigarette smoke is a significant risk factor for CVD and a main source of Cd, thus leading several studies to attempt to characterize the molecular mechanism(s) of the increased Cd-related CVD incidence [70,71,72]. Indeed, cigarette smoke, therefore Cd as well, induces vascular damage by stimulating vascular plaque inflammation and vasomotor dysfunction [73]. Five cross-sectional studies, recently reported by the National Health and Nutrition Examination Survey (NHANES) involving the U.S. population, confirmed that subjects with higher levels of either blood or urinary Cd had increased risks of peripheral artery disease, hypertension, heart failure, myocardial infarction and stroke [74,75,76,77].

Table 2 lists all significant manuscripts reporting clinical studies investigating the association of Cd, highlighting the above-discussed related findings.

#### 5.1.2. In Vitro Studies

It is well known that the genesis of atheromatous degeneration is a complex mechanism, which has determined several players including endothelium permeability. Indeed, in vitro studies characterize Cd as a pro-atherogenic factor with a cytotoxic effect in macrophages. In particular, our research group has published data demonstrating that Cd exposure can alter androgen receptor levels in Human Umbilical Vein Endothelial Cells (HUVECs) and even more importantly can stimulate pro-inflammatory signaling, strongly indicating a role for Cd in cell injury linked to endothelial damage and CVD [86]. Moreover, as reported in Table 3, Cd can also cause endothelial cell dysfunction since it alters vascular endothelial cells’ permeability, decreases nitric oxide (NO) production, inhibits endothelial cell proliferation, induces upregulation of adhesion molecules such as VCAM-1 expression level, triggers endothelial cells apoptosis and alters proinflammatory cytokines levels [87].

### 5.2. BPA and Cardiovascular Effect

Bisphenol A (BPA) is a synthetic organic compound with two phenolic groups. Since the 1960s of the past century, it is largely used for the production of polycarbonate plastics (popular for their properties including transparency, and thermal and mechanical resistance), for preparation of food containers, and for epoxy resins employed for internal protective coating of food and beverage cans. It is one of the highest-volume chemicals produced worldwide. Studies of the past two decades have, however, revealed that BPA acts as an Ed, interfering as other molecules and pollutants with hormonal pathways.

#### 5.2.1. Clinical Studies

Epidemiological studies have documented an increased risk of coronary artery disease in a healthy population exposed to BPA [78,79,92]. Further, urinary BPA levels significantly correlated with peripheral arterial alterations, independently of other known CVD risk factors [80]. An interesting meta-analysis reported that urinary levels of BPA normally found in the general population correlated with increased prevalence of hypertension, diabetes and obesity [93]. NHANES in 2003 and 2004 [78], documented that a higher concentration of urinary BPA was linked to an increased risk of self-reported CVD (myocardial infarction, angina or coronary heart disease), but not of stroke. While similar data were subsequently reported by other authors who demonstrated similar associations [81], and Casey et al. showed significant correlation between urinary BPA and coronary heart disease in another survey, results were not confirmed in subsequent evaluations [82]. Moreover, the prospective study within the EPIC–Norfolk cohort depicted a positive correlation between urinary BPA concentrations and the occurrence of coronary artery disease [79]. These data demonstrated that several cross-sectional epidemiological studies found a positive correlation between levels of urinary BPA and CVD risk factors, such as hypertension and hypercholesterolemia [94]. On the other hand, a recently published study [83] performed in a subcohort of the Spanish European Prospective Investigation into Cancer and Nutrition (EPIC) did not find a significant correlation between urinary BPA levels and the risk of incident ischemic heart diseases (IHD). The apparent contradictory results of these studies and surveys might be due most likely to different experimental designs, timing of exposure and other bias, as they might be uncontrolled or residual confounding factors, such as route of administration of these pollutants, degradation time of BPA or different exposure doses evaluated in the studies [95,96,97]. Table 2 reports all significant manuscripts reporting clinical studies investigating the association of BPA, highlighting the above-discussed related findings.

Moreover, several epidemiologic studies indicated positive associations of urinary BPA level with serum IL-6 levels in both pregnant women and adult males [84,85]. Finally, several in vivo studies showed that BPA exposure increases pro-inflammatory cytokines TNF-α and IL-6, but decreases the anti-inflammatory cytokines IL-10 and transforming growth factor-β (TGF-β) in human macrophages, strongly suggesting that BPA can trigger inflammation status likely increasing the risk of CVD (Table 1).

#### 5.2.2. In Vitro Studies

A rising number of studies indicate that exposure to environmentally significant levels of BPA might increase the susceptibility for cancer in the reproductive organs and increase body weight [98,99], but also, as mentioned earlier, increase the risk of CVD [78,81]. Thus, several in vitro studies, summarized in Table 3, focused on the characterization of the mechanism(s) by which this molecule could affect endothelial cells. One of the first studies to evaluate the potential mechanism of action of BPA on endothelial cells was conducted by Andersson and colleagues, who demonstrated that BPA increased mRNA expression of vascular endothelial growth factor receptor 2 (VEGFR-2), vascular endothelial growth factor A (VEGF-A), endothelial nitric-oxide synthase (eNOS) and connexin 43 (Cx43), and also stimulated NO production in HUVEC cells, a well-known human in vitro model of endothelial cells [88]. Furthermore, they showed that BPA also stimulated expression of phosphorylated eNOS and endothelial tube formation in HUVEC, suggesting that relevant levels of BPA might lead to proangiogenic effects in human primary endothelial cells [88].

Another study, as shown in Table 3, attempted to further characterize the molecular alterations induced by BPA exposure in vitro [89]. The authors evaluated markers of cellular oxidative stress in an experimental in vitro model of hypothalamic neurons exposed to BPA, demonstrating that BPA increased, in a time- and dose-dependent manner, the production of intracellular peroxides and mitochondrial superoxide [89]. The results of this study confirmed emerging evidence indicating that a non-institutionalized human population has higher levels of urinary BPA and high levels of oxidative stress markers leading to higher risk of CVD as well as other metabolic chronic diseases.

To further demonstrate an enhancement of inflammation induced by BPA, Song et al. demonstrated in two different experimental cellular models that BPA induced COX-2 mRNA expression, along with induction of promoter activity, suggesting a direct effect on increased transcription. Moreover, BPA treatment also increased mRNA levels of the proinflammatory cytokines TNF-α and IL-6 [90].

Since clinical findings suggested that BPA might increase the risk of ischemic heart attack and also heart-function alterations, another interesting experimental study evaluated the potential effect of BPA on electrical conduction in excised hearts. Results showed that acute BPA exposure slowed electrical conduction, highlighting a potential interfering role of BPA in heart electrophysiology and therefore suggesting that an in vivo exposure could cause or exacerbate conduction abnormalities in high-risk subjects [91].

## 6. Therapies in the Context of Eds Exposure

The damage induced by Eds leads to new opportunities in terms of pharmacological intervention but also of potential interference of Eds on the efficacy of pharmacological therapy used to decrease cardiovascular risk. In particular, these molecules can play a role as a risk factor in a gender-independent manner; however, data suggest a potential role also in term of gender specificity. As previously discussed elsewhere [100], sex steroids can significantly influence cardiovascular risk in a gender-specific manner. Thus, potential exposure to Eds will have to be taken into consideration when factor risks for CVD are analysed. Moreover, Eds might influence and interfere with positive effects of estrogen replacement therapy, which will need further evaluation.

## 7. Conclusions

In conclusion, the published studies summarized here strongly indicate that Eds can trigger human health problems by interfering with hormonal pathways, inflammatory status and immune responses in both sexes. Since it is known that sex hormones might significantly alter the immune and inflammatory responses during the atherosclerosis process, causing different disease phenotypes according to sex, present data lead to the hypothesis that Eds might interfere with cardiovascular homeostasis by interfering with these processes (see Figure 2). For instance, women respond to infection and damage by an increase in both antibody and autoantibody responses, while men respond by an increase in innate immune activation, suggesting that, despite a well-known sexual dimorphism in the incidence and complications of atherosclerosis there are few data explaining the potential mechanisms underlying gender difference as a biological variable in CVD.

A limit of this review manuscript is the lack of a larger body of evidence regarding the underlying molecular and cellular mechanisms of the complex relationship among Eds, such as Cd and BPA, and clinical conditions such as CVD. The review of the literature has indicated that further research is needed to develop valuable and beneficial intervention for preventing ageing processes often accelerated by stress factors such as pollutants and specifically Eds. Future research needs to develop further in vitro and in vivo model systems, including arterial endothelial cells from micro- and macrovascular bed. New studies are required to fully characterize all the mechanism(s) involved in the process in both genders in order to attempt to develop proper prevention strategies in a sex-dependent manner.

## Figures and Tables

**Figure 1 biomolecules-12-00044-f001:**
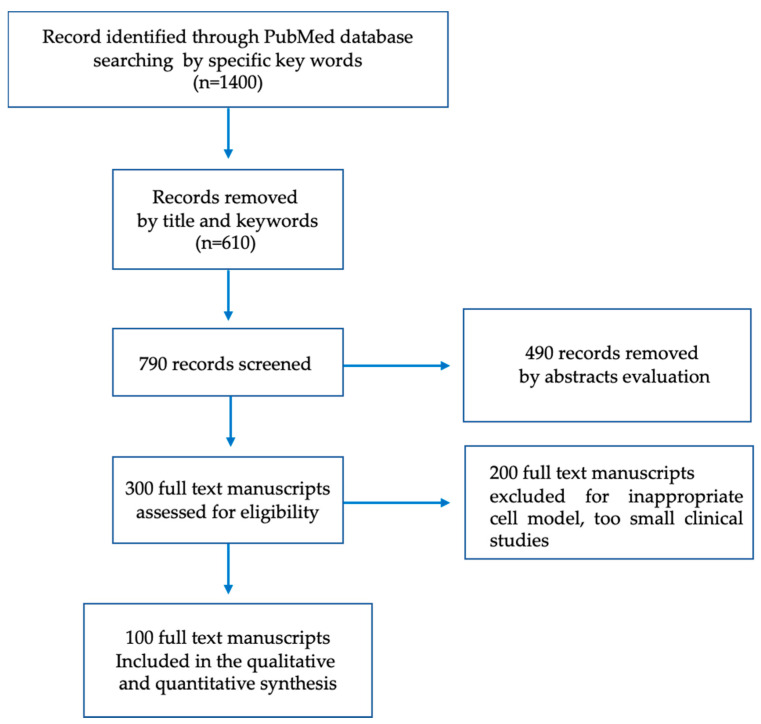
Flow diagram of the record screening and selection process leading to the inclusion of 100 studies.

**Figure 2 biomolecules-12-00044-f002:**
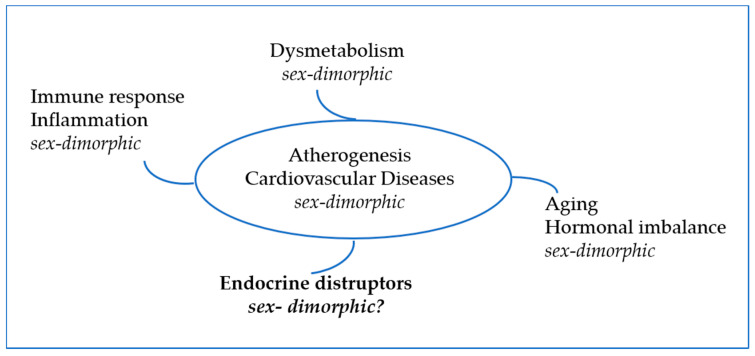
Risk factors for CVD. As the main risk factors related to CVD development exhibit sex-dimorphism, sex-dependent effects of Eds are conceivable as well.

**Table 1 biomolecules-12-00044-t001:** Cytokines involved in cardiovascular disorders.

Cytokines			
Reference	Name	Type	Effect
[20,21,22,23]	TNF-α	pro-inflammatory cytokine	decrease of insulin-mediated release of NO increase VCAM1 and ICAM1
[25,26,27,28,29]	IL6	pro-inflammatory cytokine	promotion of coagulation immune/inflammatory reaction
[30]	AT	precursor of vasoactive peptide angiotensin II	increase of blood pressure
[31]	PAI-1	serine-protease inhibitor	favouring thrombus formation over ruptured atherosclerotic plaques
[32,33,34,35]	Leptin	Adipokine	stimulation of endothelial dysfunction, atherogenesis. Induction of macrophage cholesterol ester synthesis
[38,39,40]	Resistin	Adipokine	proliferation of smooth muscle cells, increase of endothelin-I, VCAM, ICAM-1

Legend. NO: nitric oxide; ICAM-1: intercellular adhesion molecule 1; VCAM-1: vascular cell adhesion protein 1; TNF-α: Tumor necrosis factor-α; IL-6: Interleukin-6; AT: Angiotensin; PAI-1: Plasminogen activating inhibitor.

**Table 2 biomolecules-12-00044-t002:** Summary of clinical studies investigating the association between cadmium and BPA exposure.

Cadmium			
Reference	Population	Sample	Outcome
[60]	Healthy Young Females	Serum	involved in initial stages of atherosclerosis
[61]	ApoE–/– mice	Vessel sections of atherosclerotic plaques	pro-atherogenic factor
[62]	Screening population	Blood	promotion of vulnerability of carotid plaques
[63]	Korean male population	Blood	CVD mortality and carotid plaques prevalence
[64]	Swedish population	Blood	CVD mortality and carotid plaques prevalence
[65]	Swedish population	Blood	role in smoking-induced CVDs
[66]	Swedish population	Blood; urine	involved in plaques formation
[67]	patients undergoing carotid endarterectomy	Blood; FFPE tissue	increased macrophages presence
[68]	patients undergoing carotid endarterectomy	Carotid plaque	increased macrophages presence
[69]	Canadian population	Blood	increased the vulnerability of carotid plaques
[72]	Korean population	Blood	increased Cd-related CVD incidence
[74]	NHANES	Urine	increased risks of coronary heart disease
[75,76,77]	NHANES	Blood; urine	increased risks of peripheral artery disease
**BPA**			
[78]	NHANES	Urine	increased risk of self-reported CVD (myocardial infarction, angina, or coronary heart disease)
[79]	Norfolk UK	Urine	increased incident risk of coronary artery disease
[80]	NHANES	Urine	increase hypertension, independent of traditional risk factors
[81]	NHANES	Urine	positive association with CVD
[82]	NHANES	Urine	no correlation with CVD
[83]	Spanish population	Urine	no association with ischemic heart disease
[84]	Women population study	Plasma; urine	increase of IL-6, increase biomarkers of oxidative stress (including indices of oxidative DNA and lipid damage)
[85]	Male Caucasian subjects	Blood	positive association with IL-6 levels

**Table 3 biomolecules-12-00044-t003:** Summary of in vitro studies investigating the association between cadmium and BPA exposure.

Cadmium		
Reference	Cell Line/Tissue	Effects
[86]	HUVEC	stimulate pro-inflammatory signaling
**BPA**		
[88]	HUVEC	mRNA expressions increase of VEGFR-2, VEGF-A, eNOS, Cx43, stimulation of NO
[89]	GT1-7 hypothalamic neurons	increased levels of oxidative stress markers
[90]	A549 (lung cells); MDA-MB-231 (breast cancer cells)	induced COX-2, TNF-α and IL-6 mRNA expression, activation of MAPK
[91]	whole hearts (ex vivo from adult female rats)	induced a slowing of cardiac electrical conduction

Legend. Vascular endothelial growth factor receptor 2 (VEGFR-2); vascular endothelial growth factor A (VEGF-A); endothelial nitric-oxide synthase (eNOS); connexin 43 (Cx43); nitrix oxide (NO); mitogen-activated protein kinase (MAPK).

## Data Availability

All data presented are available in the manuscript text, Figure 1 and Figure 2 and Table 1, Table 2 and Table 3.

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
