# Peer review of "Environmental Contaminants Acting as Endocrine Disruptors Modulate Atherogenic Processes: New Risk Factors for Cardiovascular Diseases in Women?"

_biomolecules, 2021, doi:10.3390/biom12010044_

Round 1

Reviewer 1 Report

This is a simple minded attempt at the review of what could be an interesting topic. The description of the process of atherosclerosis is poor and not really necessary for this brief review.  There are many good reviews of the process  of atherosclerosis that could be co-opted with citation as an introduction.  Furthermore atherosclerosis is a chronic inflammation that  is localized to  selected areas of the  macrovasculature determined by hemodynamic forces. The review is agnostic  on this  dimension of the process. Being a chronic inflammation these disruptors could be influencing different stages of atherosclerosis -- a feature not easy to explore in vivo in human subjects. Have these disruptors been studied in experimental models of atherosclerosis ? These would be useful to answer some of the questions posed. The authors have ineffectively coupled the general description with the details of the effects of Cd and BPA both clinical and in vitro  on tissues and cells.  The title appears to emphasize effects on women, yet the gender differences are barely touched in the body of the review. In these reports one would like to know that appropriate controls were studied. There is no mention of the word"control" in the body of the review.

The authors should indicate if the literature is limited in its treatment of this problem. If so how can this be rectified.

Other issues 

  1. HUVEC is not  an ideal model for  arterial endothelial cells that are the target of the disrupters .
  2.  Mechanistic insights are quite limited especially bearing in mind the complexity of the atherosclerotic process
  3. Histomorphology of lesions of Cd or BPA exposed vessels could be instructive if compared with appropriate controls not so exposed.
  4. The authors have heavily focussed on endothelial cells al la Ross. But the lesion is much more complex and an experimental model could be useful in examining the involvement of other cells.
  5. Atherosclerosis is a disorder of major arteries. So what is the role of the dilation of arterioles? And periarteriolar fat?
  6. In many lines words run together eg line 98, 180, 181, etc
  7. In many places the plural has been used inappropriately -- English should be checked
  8. What is "atheromatic"== not a word in common usage in this field.

Reviewer 2 Report

  • I think that the paragraph on atheroclerotic plaque formation is redundant. The authors can shorten it and include it into the paragraph on the role of cytokines in atherosclerosis.
  • it is really interesting to evaluate the role of Eds in atherosclerotic process. The impact of therapies in the context of Eds might impact on endothelial function. This should be discussed also in relation to literature. The authors can consider the papers from Ciccone MM et al. J Cardiovasc Pharmacol Ther. 2013 Jul;18(4):354-8 and Ciccone MM et al. J Atheroscler Thromb. 2012;19(12):1061-5. Furthermore, the influence of gender modification might also impact on final outcomes (consider Ciccone MM et al. Intern Med J. 2017 Nov;47(11):1255-1262). Please discuss such points.
  • the authors should include a table gathering the main findings from literature in order to improve the readability of the text and its comprehension from readers. Specifically, the authors can consider two different tables: one related to clinical studies, the second to pre-clinical studies.
  • Please improve figure 1 as it is poorly visible. 
  •  

Reviewer 3 Report

The authors have done some interesting review work, but it has to be thoroughly reviewed to be accepted.
At present, for a review to have impact and rigor, it has to be a systematic review, where the search criteria are perfectly defined, with keywords, databases, temporality, ... as David Moher propose in this article https: //doi.org/10.1371/journal.pmed.1000097 which has more than 36,000 citations.
The review has a good structure, but in addition to making it systematic, there are some aspects to improve. For example, there is only one figure at the end of the manuscript, in conclusions at the end of the manuscript, and it should have some more supporting figure and table, which would facilitate the reading of this manuscript, for example a figure with specific risk factors in women , ...
For all this, the manuscript has to be rewritten and requires a major revision.

Author Response

Reviewer #3

The authors have done some interesting review work, but it has to be thoroughly reviewed to be accepted.
At present, for a review to have impact and rigor, it has to be a systematic review, where the search criteria are perfectly defined, with keywords, databases, temporality, ... as David Moher propose in this article https: //doi.org/10.1371/journal.pmed.1000097 which has more than 36,000 citations.
The review has a good structure, but in addition to making it systematic, there are some aspects to improve. For example, there is only one figure at the end of the manuscript, in conclusions at the end of the manuscript, and it should have some more supporting figure and table, 

Authors’ response: We thank the Reviewer for appreciating the structure of our review. Further, we thank the Reviewer for her/his careful revision and for the useful suggestions that we believe have improved our manuscript. Indeed, we carefully examined our research criteria and modified the manuscript in response to the comments, in order to satisfy the criteria of a systematic review.

Reviewer 4 Report

The aim of the paper “Environmental contaminants acting as endocrine disruptors 2 modulate atherogenic processes: new risk factors for cardiovascular diseases in women?” is the presentation of the role of pollutants which are involved in hormonal pathways leading to atherosclerosis and cardiovascular diseases, namely Endocrine Disruptors (EDs), by an extensive literature review, including authors’ previous reports.

Broad comments:

The manuscript is well organized into 6 sections, adding new data regarding the role of Bisphenol A (BPA) to their previously research concerning Cadmium (Cd) and its cardiovascular effect. The review is based on 99 reviewed articles, including nine previous articles of the authors. However, there are numerous English language errors, data are illustrated in a single image (Figure 1), which is very small and non-attractive, and considering that there are nine self-citations it is raising serious concerns.

Specific comments:

In order to match the title and the text of the manuscript, the keywords should include “women” instead of “gender” and “female”.

The sections and subsections of the manuscript are appropriate. It is recommended to introduce a table/figure to section 3. Proinflammatory cytokines effects on cardiovascular system, in order to highlight the involvement of each of them (TNF-α, IL-6, AT, PAI-1, leptin, and resistin).

Unfortunately, the single figure of the manuscript is placed in section 6. Conclusions. It is recommended to replace it in the text of the manuscript. Furthermore, an improved design of the figure would be required, as the font it is difficult to read due to minimal size and is not attractive due to lack of color and simplicity of the graphic representation.

Furthermore, the section 6. Conclusions should be reformulated, highlighting the value of the updated information provided in the text of the manuscript, which are the limits, and what is the future direction of study in this domain.

Considering that the list of references contains nine previous articles of the authors (references 4, 5, 11, 12, 14, 23, 26, 27, and 78), the authors have to consider if all these are really required or the number of self-citations is really too high (>9% of the references).

Author Response

Reviewer #4

Broad comments:

The manuscript is well organized into 6 sections, adding new data regarding the role of Bisphenol A (BPA) to their previously research concerning Cadmium (Cd) and its cardiovascular effect. The review is based on 99 reviewed articles, including nine previous articles of the authors.

Authors’ response: We thank the Reviewer for the careful and constructive evaluation of our manuscript.

However, there are numerous English language errors, data are illustrated in a single image (Figure 1), which is very small and non-attractive, and considering that there are nine self-citations it is raising serious concerns.

Authors’ response: As suggested by the Reviewer, the manuscript underwent a thorough revision correcting any errors of the English language, Moreover, in the revised manuscript you can find an improved version of Figure 1 (Figure 2, in the revised version of the manuscript) as well as  details regarding our research criteria, addressing all point raised and modifying the manuscript accordingly.

Specific comments:

In order to match the title and the text of the manuscript, the keywords should include “women” instead of “gender” and “female”.

Authors’ response: We thank the Reviewer for the suggestion, we have modified the keywords as indicated.

The sections and subsections of the manuscript are appropriate. It is recommended to introduce a table/figure to section 3. Proinflammatory cytokines effects on cardiovascular system, in order to highlight the involvement of each of them (TNF-α, IL-6, AT, PAI-1, leptin, and resistin).

Authors’ response: We thank for the comment, and we have added a Table (Table 1) to better describe the role of each cytokine as suggested.

Unfortunately, the single figure of the manuscript is placed in section 6. Conclusions. It is recommended to replace it in the text of the manuscript. Furthermore, an improved design of the figure would be required, as the font it is difficult to read due to minimal size and is not attractive due to lack of color and simplicity of the graphic representation.

Authors’ response: We thank the Reviewer for the comment, and we apologize for the scarce quality of the figure. We have provided a new figure in the revised version of the manuscript. New tables and a new figure are also included to graphically enrich the revised manuscript.

Furthermore, the section 6. Conclusions should be reformulated, highlighting the value of the updated information provided in the text of the manuscript, which are the limits, and what is the future direction of study in this domain.

Authors’ response: We thank the Reviewer for this comment. We have added a paragraph regarding the limits and, also, the future direction of the research in this field.

Considering that the list of references contains nine previous articles of the authors (references 4, 5, 11, 12, 14, 23, 26, 27, and 78), the authors have to consider if all these are really required or the number of self-citations is really too high (>9% of the references).

Authors’ response: We thank the Reviewer for the revision and for the suggestion. However, we believe that the articles we have quoted are worth mentioning since they cover different issues further described in the manuscript.

Round 2

Reviewer 1 Report

None

Reviewer 2 Report

the authors wel addressed my previous comments. The paper improved very much.

Reviewer 3 Report

Congratulations to the authors for this new version of the manuscript, that must be accepted in the journal Biomolecules.

Reviewer 4 Report

Brief summary:

The aim of the paper “Environmental contaminants acting as endocrine disruptors 2 modulate atherogenic processes: new risk factors for cardiovascular diseases in women?” is the presentation of the role of pollutants which are involved in hormonal pathways, which lead to atherosclerosis and cardiovascular diseases, namely Endocrine Disruptors, in, by an extensive literature review, including authors’ previous reports.

Broad comments:

The manuscript is well organized into 6 sections, adding new data regarding the role of Bisphenol A (BPA) to their previously research concerning Cadmium (Cd) and its cardiovascular effect. The review is based now on 100 reviewed articles, including nine previous articles of the authors. English language has been extensively revised, data are illustrated in several tables and figures, one of them being now clearer. There are still nine self-citations, which have been considered as mandatory by the authors.

Specific comments:

In order to match the title and the text of the manuscript, the keywords now include “women” instead of “gender” and “female”.

The sections and subsections of the manuscript are appropriate. Several tables and a figure have been added, according to recommendations, although one is placed in section 6 of Conclusions (now showing an improved design).

Furthermore, the section 6. Conclusions has been reformulated, highlighting the value of the updated information provided in the text of the manuscript, which are the limits, and what is the future direction of study in this domain.

The list of references contains nine previous articles of the authors, but selected by them during their search of the literature, as valuable for their review (now representing 9% of the references).